# Computed Tomography Angiography for Detection of Pulmonary Embolism in Western Australia Shows Increasing Use with Decreasing Diagnostic Yield

**DOI:** 10.3390/jcm12030980

**Published:** 2023-01-27

**Authors:** David Youens, Jenny Doust, Ninh Thi Ha, Peter O’Leary, Cameron Wright, Paul M. Parizel, Rachael Moorin

**Affiliations:** 1Health Economics and Data Analytics, School of Population Health, Curtin University, Bentley 6102, Australia; 2Australian Women and Girls’ Health Research Centre, School of Public Health, University of Queensland, St Lucia 4072, Australia; 3Division of Obstetrics and Gynaecology, Faculty of Health and Medical Sciences, The University of Western Australia, Crawley 6009, Australia; 4PathWest Laboratory Medicine, QE2 Medical Centre, Nedlands 6009, Australia; 5Fiona Stanley Fremantle Hospitals Group, Murdoch 6150, Australia; 6Medical School, Faculty of Health and Medical Sciences, The University of Western Australia, Nedlands 6009, Australia; 7School of Medicine, College of Health & Medicine, University of Tasmania, Hobart 7005, Australia; 8Department of Diagnostic & Interventional Radiology, Royal Perth Hospital, Perth 6000, Australia; 9School of Population and Global Health, The University of Western Australia, Nedlands 6009, Australia

**Keywords:** computed tomography, pulmonary embolism, overuse, data linkage

## Abstract

(1) Background: Pulmonary embolism (PE) can be fatal. Computed tomography pulmonary angiography (CTPA) can accurately diagnose PE, but it should be used only when reasonable pre-test probability exists. Overtesting with CTPA exposes patients to excess ionizing radiation and contrast media, while PE overdiagnosis leads to the treatment of small emboli unlikely to cause harm. This study assessed trends in CTPA use and diagnostic yield. We also assessed trends in PE hospitalizations and mortality to indicate PE severity. (2) Methods: Analysis of Western Australian linked administrative data for 2003–2015 including hospitalizations, emergency department (ED) attendances, and CTPA performed at hospitals. Age-sex standardized trends were calculated for CTPA use, PE hospitalizations, and mortality (as a proxy for severity). Logistic regression assessed diagnostic yield of CTPA following unplanned ED presentations. (3) Results: CTPA use increased from 3.3 per 10,000 person-years in 2003 (95% CI 3.0–3.6) to 17.1 per 10,000 person-years (16.5–17.7) in 2015. Diagnostic yield of CTPA increased from 12.7% in 2003 to 17.4% in 2005, declining to 12.2% in 2015 (*p* = 0.049). PE hospitalizations increased from 3.8 per 10,000 (3.5–4.1) in 2003 to 5.2 per 10,000 (4.8–5.5) in 2015. Mortality remained constant at 0.50 per 10,000 (0.39–0.62) in 2003 and 0.42 per 10,000 (0.32–0.51) in 2015. (4) Conclusions: CTPA increased from 2003 to 2015, while diagnostic yield decreased, potentially indicating overtesting. PE mortality remained constant despite increasing hospitalizations, likely indicating a higher proportion of less severe cases. As treatment can be harmful, this could represent overdiagnosis.

## 1. Introduction

Pulmonary embolism (PE) occurs when an embolus, usually originating from lower limb deep veins, obstructs the pulmonary arterial system [1]. PE can be a life-threatening condition [2]. One of the challenges in diagnosing PE is the range of non-specific symptoms, such as chest pain and dyspnea, which have a wide differential diagnosis [1,2]. As these symptoms appear in many less serious conditions, most patients with these symptoms will not have PE [2].

Since the introduction of multi-row detector CT scanners in the 1990s, computed tomography pulmonary angiography (CTPA) has overtaken ventilation/perfusion (V/Q) scanning in the diagnosis of PE [3]. CTPA is the gold standard for diagnosing PE due to its high sensitivity and specificity [1,4]. However, CTPA is not without risks for the patient, such as exposure to ionizing radiation, injury from iodinated intravenous contrast agent injection, and increased use of additional investigations to resolve incidental imaging findings or false positives. These considerations should always be taken into account, especially when ordering a CTPA for a patient at low risk of PE. Other diagnostic tools are recommended to rule out PE before progressing to CTPA for a definitive diagnosis; these include clinical decision rules based on X-ray findings, patient risk factors, such as Wells scores and pulmonary embolism rule-out criteria, and D-dimer testing [1,5]. Recent literature indicates that D-dimer testing was used in less than half of patients with CTPA and the yield rate for PE was higher in centers with higher D-dimer usage [6].

Nevertheless, there is evidence that CTPA is substantially overused [6]. Reasons for CTPA overuse may include the availability of CT scanners [4], a lack of understanding of the potential consequences of excessive imaging, and clinician and patient concerns about missing PEs [7]. Emergency department providers are the main drivers of ordering CTPA, and their decisions appear to be based on personal experience and/or gestalt-based heuristics rather than evidence-based algorithms or D-dimer values [8].

Overuse of CT is associated with potentially adverse outcomes. CTPA exposes patients to ionizing radiation (5 millisieverts (mSv) to 10 mSv) and potential risk of nephrotoxicity from iodinated contrast media (9–21 g of iodine) [1,9]. Radiation exposure is especially of concern in vulnerable and high-risk population groups, such as children/adolescents, young women, and pregnant women [10]. The discussion about contrast-induced nephrotoxicity remains somewhat controversial, although it is clear that caution is warranted in patients with renal risk factors such as acute and chronic nephropathy as well as those in whom renal function is unknown [11]. Furthermore, PE treatments carry significant risks. Treatment involves anticoagulation [12], which increases bleeding risk [5]. If CT scanning results in the detection of small, non-clinically significant PEs, the balance of benefit and harm resulting from anticoagulation becomes less favorable. Finally, increased testing and decreasing diagnostic yields can result in wasted resources and higher costs [5] in relation to both the costs of the imaging itself and the costs of treatment or follow-up investigations resulting from the diagnosis of clinically non-significant findings.

These potential harms fall under two related concepts: overtesting and overdiagnosis. Overtesting refers to the use of non-recommended screening and diagnostic tests in asymptomatic patients or more testing than necessary to obtain a diagnosis [13], whereas overdiagnosis refers to a person being diagnosed with a condition where the diagnosis does not produce a net benefit [14].

In the United States, the Choosing Wisely campaign published recommendations against using CTPA as a first-choice investigation in adults with low risk of PE and a negative D-dimer test in 2014 [15], with similar recommendations in other countries since then [16,17]. Similarly, the Western Australian (WA) Department of Health (the State in which the current study was conducted) has produced ‘Diagnostic Imaging Pathways’ outlining best practice in PE diagnosis, including the imaging options recommended for cohorts with different pre-test probabilities of PE and the associated effective doses [18]. These types of diagnostic protocols have been demonstrated to improve the diagnostic yield of CTPA [19,20]. Investments in infrastructure have also aimed to reduce excessive imaging, for example, the introduction of a Picture Archiving and Communications System (PACS) in 2009 [21].

There is evidence that the introduction of CTPA in the United States was associated with an increase in PE prevalence recorded but no change in PE mortality and an increase in treatment-related bleeding [22]. However, diagnostic yield is known to vary substantially between countries [23]. Research from WA found that a 65% increase in CTPA scanning from 2002 to 2010 was accompanied by a 45% increase in PE hospitalizations [24]. However, the death rate did not change, suggesting CTPA overuse [24]. In contrast, other Australian research reported substantial differences in CTPA usage and diagnostic yield between EDs, although in this case, increased use did not increase the proportion of small PEs detected, suggesting that increased use did not lead to overdiagnosis [25]. The most recent WA data reported dramatic increases in CTPA use up until 2010 [24]. More up-to-date analyses are warranted to assess changes in CTPA use and diagnostic yield since that time. Australia has a high number of CT scanners in comparison to other OECD countries [26] and a fee-for-service health system, both factors known to increase the risk of overtesting [27].

The aims of the current work are therefore: (1) to assess potential overtesting by describing trends in CTPA utilization and diagnostic yield in WA from 2003 to 2015, and (2) to assess potential overdiagnosis by describing mortality outcomes as a proxy marker of PE severity.

## 2. Materials and Methods

This was a retrospective observational cohort study using individual-level linked administrative health data. Data extraction and probabilistic linkage were undertaken by the Data Linkage Branch of the WA Department of Health [28], which reports low rates of false positives and false negatives of 0.11% each [29]. The findings are reported following the RECORD checklist [30].

### 2.1. Cohort and Data

The overall cohort was made up of any WA resident aged 18+ who was discharged from any WA hospital for a condition other than pregnancy-related conditions, presented to any public emergency department (ED), or had a WA Health Picture Archival Communication System (PACS) record for a CT scan recorded from 2003 to 2015. The following datasets were available:(1)Hospital morbidity data collection (HMDC) records from 2003 to 2015 for all discharges from any WA hospital (public or private) for all conditions excluding pregnancy.(2)Emergency Department Data Collection (EDDC) records 2003–2015 for all WA ED presentations.(3)PACS data for all CT scans in 2003–2015 undertaken in all WA public tertiary hospitals and some public secondary hospitals. Note that only secondary hospitals operating their own radiology departments (i.e., do not contract out radiology services) are included in the PACS data. Two secondary hospitals, Bentley Hospital and Armadale Health Service, are therefore excluded from the PACS. These hospitals account for approximately 6% of public hospital inpatient admissions in WA [31].(4)WA death registrations from 2003 to 2015.

Age- and sex-specific population estimates for the WA population aged 20+ were obtained from the Australian Bureau of Statistics to allow age-sex standardization [32].

### 2.2. Sub-Cohorts for Each Analysis

The study involved multiple cohorts that were analyzed separately. Table 1 provides an overview of the cohorts and outcomes for each aim.

The first cohort, used to assess population trends, was made up of the 2016 WA population aged 20+.

The second cohort was based in the ED, consisting of those with a CTPA examination performed the day of or day after ED presentation. This set of CTPA examinations formed the denominator in estimating the diagnostic yield. Since datasets did not indicate multiple services related to the same episode of care (e.g., an ED presentation and related PACS record), these were assumed to be related where dates aligned.

### 2.3. Outcomes

Outcomes summarized in Table 1 were used as dependent variables in regression models or as numerators in rates.

The first outcome, applied to the WA population cohort, was the number of CTPAs performed per year from the PACS data. As a comparator, the number of other CT angiography examinations was also assessed; if trends in CTPA use resulted primarily from scanner availability then similar trends may be expected in other forms of CT angiography. Other CT angiography excluded stent work-ups and a subset of thoracic aorta and head scans that appeared to be coded inconsistently throughout the study period.

The second outcome, applied to the ED cohort, was a binary variable indicating hospital admission with PE (ICD-10-AM codes I26.0 and I26.9) as the primary or secondary diagnosis the day of or after the CTPA. The restriction to primary or secondary diagnoses was because other diagnosis fields are more likely to reflect a pre-existing comorbidity or a condition developing during hospitalization rather than the reason for the current admission.

The final outcomes, applied to the WA population cohort, were numbers of PE-related hospitalizations (as described earlier) per year and the numbers of deaths during these hospitalizations, which were indicated by the Mode of Separation on HMDC records.

### 2.4. Analysis

Aim 1a was to assess overtesting by describing trends in overall CTPA utilization. Age-standardized rates were produced to assess trends over time. Age- and sex-specific counts (5-year age groups) of the numerators (use of CTPA and other CT angiography) were tabulated. These counts were applied to population estimates for each age-sex category as of July each year to produce annual age- and sex-specific rates. The relative size of each category in 2016 was applied to all years to standardize the 2016 population.

Aim 1b was to assess overtesting based on diagnostic yield of CTPA in tertiary hospital EDs. For this aim, multivariable logistic regression was used with hospital admission for PE applied as the dependent variable. The assumption underpinning this outcome was that any PE diagnosed in the ED would result in admission. Year was included as a factor variable and adjusted probabilities of hospitalization produced for each year using the margins command in Stata 14 [33]. Significance in trends over time was assessed using the postestimation contrast command in Stata, providing tests for linear, quadratic, and higher-order trends. Clustered robust standard errors were used as one patient could contribute multiple ED records.

Aim 2 was to assess overdiagnosis by describing population trends in PE hospitalization and in-hospital mortality. These outcomes were applied as numerators to the WA population data, following the procedure described for aim 1a. This analysis was complicated by changes in thromboembolism treatment throughout the study period since improved treatment would impact death rates. The first novel oral anti-coagulant (NOAC) was approved for use in Australia in 2008, with the number of NOACs available and range of approved indications increasing through to 2015 [34]. As a robustness test, this analysis was repeated with cerebrovascular hospitalization (ICD-10-AM I60-I69) and death using cerebrovascular hospitalization as the outcome since the introduction of NOACs would also impact these conditions.

### 2.5. Covariates

For aim 1b, covariates predominantly came from the EDDC. The available socio-demographic information included age (continuous), sex, and postcode, which allowed for determination of area-level socioeconomic status (using the Socio-Economic Index for Areas Index of Relative Social Disadvantage [SEIFA-IRSD]) [35] and remoteness (using the Accessibility/Remoteness Index of Australia [ARIA]) [36]. Covariates related to ED presentation included triage code, arrival type (private transport, ambulance, other) and referral source (self, general practitioner/clinic, other). Comorbidity was determined from the HMDC using the Multipurpose Australian Comorbidity Scoring System (MACSS) [37], applied as the number of 17 MACSS conditions recorded on any hospitalization record in the prior 5 years, updated annually. Missing data were coded as an additional category to prevent data loss.

## 3. Results

### 3.1. Trends in Use of CTPA (Aim 1a)

Trends in CTPA use were assessed in a population of approximately 1.9 million (Table 1), evenly split between males and females with a mean age of 47 years. Figure 1A demonstrates substantial increases in CTPA use. Usage increased from 3.3 CTPAs per 10,000 person-years in 2003 (95% CI 3.0–3.6) to 17.1 per 10,000 person-years in 2015 (CI 16.5–17.7), with CIs indicating significant increases. Increases in other CT angiography also occurred, from 6.3 per 10,000 person-years in 2003 (95% CI 5.9–6.8) to 22.0 per 10,000 person-years in 2015 (CI 21.3–22.7).

### 3.2. Diagnostic Yield of CTPA Performed Following ED Presentation (Aim 1b)

There were 11,698 occasions of unplanned ED attendance with associated CTPA among 10,707 people, with 1525 associated PE diagnoses (Table 1). This majority of this cohort was female (54.2%) and the mean age was 61.5 years. Figure 1B suggests that diagnostic yield for CTPA after unplanned ED presentation increased early in the study period before declining. Yield increased from 12.7% (95% CI 7.4–18.0) in 2003 to 17.4% (13.1–21.6) in 2005, before decreasing to 12.2% (10.6–13.7) in 2015 (model outputs, including covariates, in Appendix A). There was a linear trend in diagnostic yield over time (Chi2 (1) = 3.86, *p* = 0.049).

### 3.3. Trends in PE Hospitalisation and Death during PE Hospitalisation (Aim 2)

Figure 2 suggests that the age-sex standardized rate of PE hospitalizations increased, from 3.8 per 10,000 person-years (95% CI 3.5–4.1) in 2003 to 5.2 per 10,000 person-years (4.8–5.5) in 2015. The rate of in-hospital deaths, however, remained constant, changing from 0.50 (CI 0.39–0.62) per 10,000 in 2003 to 0.42 (0.32–0.51) per 10,000 in 2015.

A contrast was observed with cerebrovascular conditions (Appendix A) for which slight declines were observed in rates of cerebrovascular hospitalizations (15.6 per 10,000 [CI 15.0–16.3] in 2003 to 14.9 per 10,000 [CI 14.4–15.5] in 2015) and in-hospital deaths (4.6 per 10,000 [CI 4.3–5.0] in 2003 to 4.0 per 10,000 [CI 3.7–4.2] in 2015) for these conditions.

## 4. Discussion

CTPA use increased from 2005 to 2015 while diagnostic yield amongst ED patients decreased. The PE hospitalization rate also increased, while the outcome of PE mortality remained constant. Although this study cannot prove that overuse occurred, increasing CTPA use with decreasing diagnostic yield suggests an increased likelihood of overtesting over time, while the combination of increasing hospitalizations and constant mortality may reflect an increase in the diagnosis of clinically non-significant emboli, i.e., overdiagnosis.

These results suggest a continuation of trends previously described in WA by Segard et al. [24], who found that increased use of CTPA from 2002 to 2010 coincided with an increase in D-dimer testing, increases in admissions for PE, decreasing diagnostic yield, and constant mortality. The authors hypothesized that indiscriminate D-dimer testing among patients with low pre-test probability was contributing to increased CTPA referrals [1]. The trends in CTPA use, PE hospitalization, and PE-related death observed here were consistent with those reported by Segard et al. and suggest a continuation of the same trends; meanwhile similar trends to those observed in our study have been reported in other settings [22].

A previous evaluation of ED-ordered CTPA across Australasian EDs reported a yield of 14.6% in 2014 [25], compared to 12.9% in this study, with the difference possibly due to the different set of hospitals included. The diagnostic yield reported here is higher than rates reported in North America [12,19] but below rates reported in Europe [23]. A challenge in interpretation is the lack of any target for diagnostic yield that balances the harm of undiagnosed emboli against the harm of overuse. The Royal College of Cardiologists in the UK has suggested a target of 15% or above for CTPA, though this is a benchmark derived from observations in that country, rather than reflective of a balance of benefit and harm [38]. Australian researchers suggested that a target of 10% would be too low, although they did not suggest an appropriate target [25].

Alongside the radiation concerns associated with decreased diagnostic yield, increased use of CTPA may have implications for patients in terms of incidental findings. A recent multi-center study in Canada reported that among 1708 CTPAs performed at tertiary EDs, almost as many returned incidental findings (13.1%) as resulted in a PE diagnosis (13.6%) [39]. Of the 223 incidental findings, only 11.7% were clinically significant (i.e., resulted in a diagnosis of cancer or other condition requiring treatment) and there were 17 additional CT scans per each significant incidental finding. Similarly, the potential detection of non-significant PEs may expose patients to further risk in terms of adverse treatment effects. A recent meta-analysis of over 15,000 patients with subsegmental PE diagnosed via CTPA reported that 90-day embolism recurrence did not differ between patients treated and untreated with anticoagulation, while 8% of those in the anticoagulation group reported bleeding (data not available for untreated group) [40]. Other researchers have suggested that in patients with subsegmental PE, the risk of recurrent thromboembolism in the short-term may be lower than the risk of adverse events from anticoagulation in certain groups [41].

In addition to the patient risks associated with excessive CT use and reduced diagnostic yield, this can also present a substantial cost to the health system. At an approximate cost to the government of AUD 598 [42] (2015 cost adjusted to 2022 values, AUD 1= USD 0.69), the diagnostic yield of 12.2% reported in 2015 translates to a cost of AUD 4903 per PE detected compared to AUD 3438 in 2005 when diagnostic yield was at its highest. There are also costs associated with increasing rates of PE admission. In Australia, minor complexity PE hospitalizations cost the government on average AUD 5518 [43]. Thus, the increase in PE hospitalizations from 3.8 to 5.2 per 10,000 person-years at risk throughout this study translates to an increased cost of AUD 7725 per 10,000 person-years at risk. If this increase is due in part to overdiagnosis of non-significant emboli, this represents a substantial cost at the system level.

Trends observed in CTPA use observed in this study were generally consistent with other types of CT angiography, which may indicate that increased scanner availability could be an influential factor. Even if drivers of increased usage were similar across types of CT angiography, consequences may differ considering the differences in radiation dose between types. It is unclear the extent to which treatment changes influenced trends observed, alongside changes in diagnostic testing. There were decreases in both hospitalizations and mortality for other cerebrovascular conditions; both trends may have resulted from the introduction of NOACs [34]. In the case of PE, the same medications might have influenced the mortality rate, although the substantial increase in hospitalization rates observed is unlikely to be explained by improvements in treatment.

Recommendations have been made to minimize overuse, including incentivizing appropriate pre-test probability assessment, dissociating payments from test ordering, and incorporating these issues into clinical guidelines [44]. Researchers have advocated for public debate on overdiagnosis [14] and for clinicians to be provided with clearer information on reducing the absolute risk associated with screening programs [45]. In relation to PE, evidence-based clinical decision support systems (CDS) have been shown to improve diagnostic yield [19,20], and WA Health provides these in relation to CTPA [18] although they may not always be used. In patients suspected of PE, the obvious way forward is to embed a pre-test probability assessment in the ED workflow using electronic health records to present CDS related to PE diagnostic strategies to the ED physician [46]. Such systems have been shown to decrease advanced imaging use, improve the diagnostic yield of CTPA, and reduce non-adherence to pre-test probability assessment [46]. Similar international research has suggested that mandating the recording of Wells scores and D-dimer values on CTPA request forms increases diagnostic yield [47], although such policies may oversimplify the role of clinical judgement. In WA hospitals, CTPA would be requested by a physician, with the radiology department making the ultimate decision as to whether the scan is performed. In some settings, this is supported through radiology registrars contactable for discussion of imaging requests, with the aim of rationalizing requests. In addition, patient anxiety regarding diagnostic uncertainty may influence decision-making [48]. Therefore, there are multiple avenues to target interventions, including the requesting physician, the radiology/imaging department, and the patient.

### Strengths and Limitations

Data used here cover all tertiary hospitals and EDs in Western Australia, as opposed to some studies that focus on a single institution. The availability of both hospital and ED data provided a more comprehensive overview of the issue than research in a single setting.

Several study limitations should be noted. The primary limitation of this work is the lack of detailed clinical information available. The assessment of PE severity relied on mortality as a proxy indicator as we lacked more direct measures, such as emboli size. The potential limitation of this approach is that the stable PE mortality rate (in contrast with increasing hospitalizations) may reflect both changing diagnostic practices (the focus of this paper) and improved management of PE. We have attempted to address this within the confines of the administrative data by incorporating trends in hospitalization for and mortality from cerebrovascular conditions, as mortality from both cerebrovascular conditions and PE may have improved following the introduction of NOACs. Cerebrovascular conditions do not show a divergence between hospitalizations and deaths, as is the case for PE, thereby increasing our confidence that this treatment change did not wholly drive the trends observed in PE, although other changes in management may have occurred.

This work does not comment on radiation dosages, which may change over time with changes in scanning equipment and practices and are not observable in our data. Even if radiation dosages have reduced, other problems discussed in terms of overtesting and overdiagnosis, such as anticoagulation therapy for small Pes and resource allocation issues, remain relevant. Radiation safety is guided by the “as low as reasonably achievable” (the ALARA) principle [49] and any overuse violates this principle, especially in vulnerable individuals such as children, adolescents, young women, and (potentially) pregnant women. The same should hold true for the use of iodinated contrast media in patients with pre-existing renal impairment, critically ill patients, or elderly patients with unknown renal function.

We did not have access to CT data from two secondary hospitals, although given the small proportion of inpatient admissions in WA made to these hospitals, it is unlikely that this exclusion could substantially alter the trends reported.

Increases in CTPA use may reflect an increase in imaging for PE in general, or substitution of CTPA for V/Q scanning. Our data did not include V/Q scans. However, we assessed population trends in PE hospitalization and mortality regardless of diagnostic technique and the observed trends were consistent with overdiagnosis, suggesting that the CTPA trends likely reflect increased scanning rather than substitution.

Interpretation of CTPA data is susceptible to misdiagnosis. There is increasing evidence that deep learning models can detect PE on CTPA with satisfactory sensitivity and an acceptable number of false positive cases [50]. Given the timeframe covered in this study (2003–2015), radiologic interpretation of CTPA studies was performed without access to deep learning algorithms. As artificial intelligence tools are increasingly being implemented in radiology departments, future studies assessing over- or underdiagnosis of PE on CTPA may need to take into account the strengths and limitations of deep learning models.

Finally, we had access to data covering a period to the end of 2015; future research should assess these issues through the subsequent period.

## 5. Conclusions

From 2003 to 2015, there was a substantial increase in CTPA use in Western Australia, accompanied by decreased diagnostic yield, increased PE-related hospitalizations, and a decreased proportion of those hospitalizations resulting in death. These trends are consistent with overtesting and overdiagnosis, although interpretation is hampered by a lack of diagnostic yield target. To limit the potential for overtesting and overdiagnosis, the use of evidence-based clinical decision support or diagnostic decision pathways should be encouraged. Future study should assess the impact of the Australian Choosing Wisely guidelines, which were introduced since the end of this study.

## Figures and Tables

**Figure 1 jcm-12-00980-f001:**
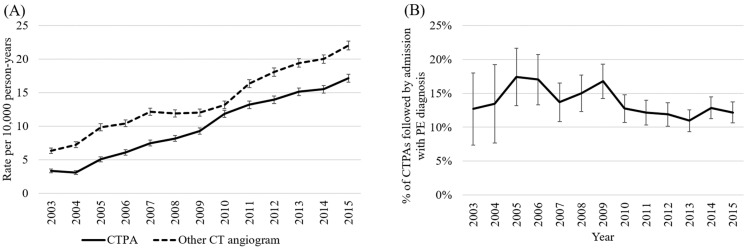
(**A**) Rates of CTPA and other CT angiography; (**B**) Diagnostic yield of emergency department CTPA. Legend: Rates of CTPA and other CT angiography adjusted for age and sex. Other CT angiography in part A excludes records relating to stent work-ups and a subset of thoracic aorta and head scans with apparent coding changes during the study period. Diagnostic yield (part B) reflects the adjusted % of CTPAs performed during an unplanned ED presentation that were followed by admission with PE as the primary/secondary diagnosis. Model adjusted for sex, socio-economic status, remoteness, triage code, arrival type, and referral source. Abbreviations: CTPA, computed tomography pulmonary angiography; ED, emergency department; PE, pulmonary embolism.

**Figure 2 jcm-12-00980-f002:**
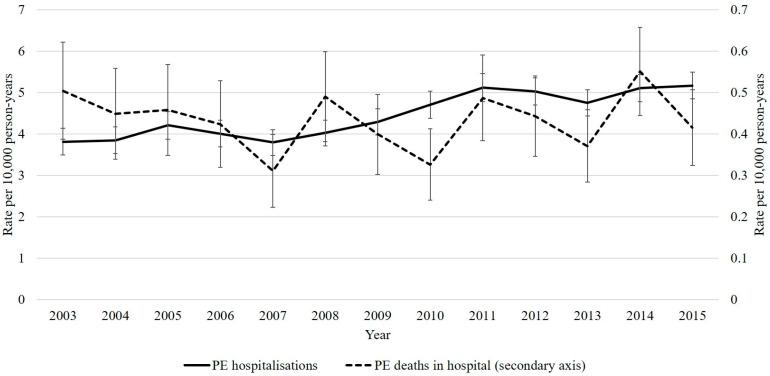
Age-sex standardized rate of PE hospitalizations and rates of death during PE-related hospitalizations, by year. Legend: PE hospitalization defined as a hospitalization with PE as the primary or secondary diagnosis. Death during PE-related hospitalization based on the mode of separation for this set of hospitals. Age-sex standardized to the 2016 Western Australian population aged 20+. Abbreviations: PE, pulmonary embolism.

**Table 1 jcm-12-00980-t001:** Details of the cohorts and outcomes for each analysis, with basic descriptive statistics for each.

Objective	1a: Assess Overtesting by Describing Trends in Overall CTPA Utilization	1b: Assess Overtesting by Assessing Diagnostic Yield of CTPA Requested by ED Physicians in Tertiary Hospitals	2: Assess Overdiagnosis by Describing Population Trends in PE Hospitalization and In-Hospital Mortality
Cohort	WA population aged 20+ 2003–2016	Those with CTPA performed day of or after an ED presentation at a tertiary hospital	2016 WA population aged 20+ (trends in hospitalization)
Outcome	Annual count of CTPA examinations	Subsequent hospital episode with PE as primary or secondary diagnosis	PE hospitalization and death during PE hospitalization
Analysis	Age-sex standardized rates	Logistic regression	Age-sex standardized rates
N	Approx. 1.9 million ^1^	11,968 CTPAs among 10,707 people	Approx. 1.9 million ^1^
Sex ^1^	Male 49.8% Female 50.2%	Male 5362 (45.8%) Female 6336 (54.2%)	Male 49.8% Female 50.2%
Mean age	46.9	61.5	46.9
Proportion with outcome		1525 (13.0%)	

^1^ Figures for 2016 provided. Abbreviations: CTPA, computed tomography pulmonary angiography; ED, emergency department; PE, pulmonary embolism.

## Data Availability

Datasets used for this article are unable to be made publicly available due to privacy and ethical considerations.

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
