# Peer review of "Computed Tomography Angiography for Detection of Pulmonary Embolism in Western Australia Shows Increasing Use with Decreasing Diagnostic Yield"

_jcm, 2023, doi:10.3390/jcm12030980_

Round 1
Reviewer 1 Report
This is an interesting study that further contributes to the literature suggesting an increase in the utilization of CT-PA, with accompanying reduction in the yield and potentially little or no effect on patient outcomes. There are a couple of minor revisions that would increase the quality of the manuscript.
1. Would it be possible to calculate the cost of the CT-PA? It would be interesting to see the radiology cost associated with the diagnosis of 1 PE over time.
2. Is this possible that the stability of the mortality rate suggests an improvement in the management of PE? This will need to be stated in the discussion, especially since there are no data on the severity of the PE.
3. To my understanding, not every hospital in WA uses the public PACS system. What % of hospitals are not part of the public PACS? What % of the population do they cover?
Reviewer 2 Report
The authors present data of a retrospective study evaluating the temporal trends in the utilization of CT pulmonary angiography and the potential implications on pulmonary embolism diagnosis in Western Australia. This historical data is important to analyze as it allows us to better understand trends in the use of technology in medicine and the potential negative implications that may occur from over utilization of technology, resulting in "over testing and over diagnosing" and potential adverse patient outcomes. Please see my comments below.
Consider revising the title of your paper to make it clearer - "in Western Australia" is repeated and leads to confusion. I would also state that you are evaluating the use of CT angiography with a focus on pulmonary embolism.
Is there data on healthcare cost that can be attributed to increasing CTA utilization or PE hospitalization? I think this would have an even greater impact on the readers of the manuscript.
Is there data on other potential negative consequences of overdiagnosis of PE such as increase in bleeding rates from increase in treatment dose anticoagulation?
Have you considered evaluating for other potential downstream effects of excessive CTA use such as incidental pulmonary nodule findings?
